# Feasibility Study of Activated Sludge/Contact Aeration Combined System Treating Oil-Containing Domestic Sewage

**DOI:** 10.3390/ijerph17020544

**Published:** 2020-01-15

**Authors:** Chih-Kuei Chen, Hung-Chih Liang, Shang-Lien Lo

**Affiliations:** 1Department of Evironmental Engineering, National Ilan University, Yilan 260, Taiwan; cgg@ms22.hinet.net; 2Graduate Institute of Environmental Engineering, National Taiwan University, Taipei 106, Taiwan; sllo@ntu.edu.tw

**Keywords:** activated sludge/contact aeration, oil and grease, true color, SEM, DGGE, biological phase

## Abstract

Both activated sludge/contact aeration (AS/CA) and AS-only systems for treating oil-containing domestic sewage were tested. The results of these tests indicate that the oil and grease removal ratios of the AS/CA system exceeded those of the AS-only system. When the influent oil and grease concentration reached 60 mg/L, the effluent concentration of the AS system was 13 mg/L, which exceed 10 mg/L, the Taiwan Effluent Standard for oil and grease. However, in the AS/CA system, the effluent oil and grease concentration was 8 mg/L, which was below the required standard. This study analyzes the biological phases of the AS-only system and the combined AS/CA system using a scanning electronic microscope and a denatured gradient gel electrophoresis method when the inflow concentration of oil and grease is increased to 120 mg/L. The results of the Chemical Oxygen Demand (COD) experiment reveal that the AS/CA system is affected less than the AS system, and the COD removal rate of the AS/CA system is maintained at 81%, which exceeds that (61.5%) of the AS-only system. The analytical results thus obtained suggest that both the amounts of biological phase and the biomass in the combined AS/CA system exceed those of an AS-only system.

## 1. Introduction

In the last decade, a breakthrough in molecular biotechnology has enabled the extraction of DNA of microorganisms using the polymerase chain reaction (PCR) technique. The quantity of the DNA of microorganisms can increase by the PCR technique, and the DNAs in different microorganisms can be separated by the denatured gradient gel glectrophoresis (DGGE) technique, making the analysis of the biodiversity of those microorganisms feasible [1]. The method is based on the fact that each bacterium has DNA; however, its RNSs are unstable and difficult to measure. Therefore, by extracting 16S-rDNA and knowing that 16S-rDNA will varies with the species of bacteria, the specifies of bacteria and the quantity of each species can be detected [2].

Although traditional biological treatment has the advantage of low operation and maintenance costs, it requires a large place for facilities, and the activated sludge (AS) method can be utilized to treat up to only 30 mg/L of oil and grease [3]. Reduction of oil and grease may be due to biological growth that destroys the emulsifying agent, which has sufficient adsorptive power to hold oil and grease for its oxidation [4]. In contrast, the activated sludge/contact aeration (AS/CA) system combines the suspended microbes and the attached microbes to provide a long food chain and complex biophase. This is to enhance the removal of COD, oil, grease, and true color, and would require a smaller land area. This makes the system easier to operate, more stable, and more tolerant of spike loading than the conventional biological treatment process [5]. This combined system has the advantages of both the AS system and the CA system and eliminates the drawbacks of these two individual systems; it improves upon the conventional biological system, which cannot remove oil and grease effectively and has a low treatment efficiency.

The AS/CA system supports a higher bacteria growth density than does the AS system, and the microbial groups differ. These observations demonstrate that the AS/CA system contains more diverse microbes, possibly explaining its higher treatment efficiency. When 120 mg/L of oil and grease are added, the number of the transverse stripes in each reaction tank increases slightly, indicating that the size of the microbial group in each reaction tank is increased. This increase in the AS/CA system exceeds that in the AS system.

This study verifies the effectiveness of the developed AS/CA system in treating oil and grease by comparing the treatment of synthetic domestic sewage by an AS system with that by an AS/CA system [6]. Domestic sewage is a type of raw water that can be treated easily in sewage treatment facilities, and the synthetic sewage for treatment in laboratories can be utilized to examine sewage treatment using simple methods. It will be easier to do so in laboratories, since laboratories allow us to increase the constant quantity of oil and grease, maintain a stable quality of inflow water, and provide controllability of the whole sewage treatment system [7].

## 2. Materials and Methods

### 2.1. Materials

Table 1 presents the Composition of Synthetic Domestic Sewage in a small model factory. The concentrations of glucose, iron chloride, nutritious beef juice, ammonium chloride, and sodium sulfite of the synthetic domestic sewage were 400, 400, 49, 49, and 38 mg/L, respectively. The average concentrations of COD and Biochemical Oxygen Demand (BOD) of the synthetic domestic sewage were 420 and 213 mg/L, respectively.

Table 2 displays Composition of Wei Lih fragrance oil in a small model factory used in this study. For instance, the major compounds in the oil include oleic acid (48.8–55.3%), palmitic acid (18.1–21.0%), and linoleic acid (13.2–18.3%).

Table 3 displays the composition and properties of an emulsifier in the small model factory. The composition of the lauric acid used in this study was about 50%, with a balance of primarily myristic, palmitic, and setaric acids. The density of the emulsifier at 25 °C was approximately 1.095 g/mL.

Table 4 displays specifications of a biological contact filter in the small model factory. The density, specific surface area, and porosity of the biological contact filter in the model factory were 76 kg/m^3^, 274 m^2^/m^3^, and 91%, respectively.

Table 5 presents the size and effective volume of each tank in the small model factory. The effective volumes of the activated sludge, contact aeration, sediment tank square upper half, and conical lower half tanks were 4.50, 4.50, 2.25, and 0.58 L, respectively.

Figure 1 shows the flowchart of the small model factory with the AS and AS/CA Systems. The pilot plant of the AS/CA system included one activated sludge tank, one contact aeration tank, one settling tank, two metering pumps, two sludge recycling pumps, four blowers, 4.5 L of biological contact filters, one DO meter, one pH meter, one ORP meter, and one MLSS meter. The AS/CA system was converted to single AS system by clearing away biological contact filters of the contact aeration tank. The AS/CA system was converted to single CA system by adding biological contact filters into the activated sludge tank. These two systems, i.e., the single AS and AS/CA system, were under the same operating conditions, described as follows: (1) microbial acclimation period; (2) stationary period; (3) hydraulic retention time (HRT); (4) sludge recycle rate; and (5) influent substrates.

### 2.2. Experimental Setup

The controlled conditions of the small model factory were summarized as follows: (1) inflow rate was 18.75 mL/min, (2) the HRT of the biological aeration tank was 8 h, (3) the HRT of the activated sludge tank was 4 h, (4) the HRT of the contact aeration tank was 4 h, (5) the HRT of the sedimentation tank was 2.5 h, (6) the sludge return ratio was 30%, (7) the sludge retention time (SRT) was 15 days, and (8) the filling ratio of the biological contact filter in the CA tank was about 50%. In this study, the AS/CA system was created by adding biological contact filters to the back section of an activated sludge tank. The system contained both suspended and fixed microorganisms.

After the two small model factories with the AS and AS/CA systems were established and clear water trials were carried out for one week, the model factory started to fill up with synthetic domestic sewage and 2L of concentrated biological sludge from the sewage plant of the Grand Hotel, Taipei, and the experiment entered its first-stage trial period (of around 14 days). The removal rate of COD was used to determine the duration in days of the formal trial period, the cultivation period, and the stable period. After the system had stabilized, the water quality indices were determined, and the AS system was compared with the AS/CA system. Finally, various concentrations of oil and grease were added, and the emulsifier was added at a concentration of one quarter of each of these concentrations of the oil and grease. The difference between the maximum tolerable concentration of oil and grease of the AS system was compared with that of the AS/CA system.

### 2.3. Analytical Methods

The analytical methods were based on the standard methods for the examination of water and wastewater. For the water quality analysis to alter operation criteria after the system stabilized, water samples were taken from the equilibration tank and from the discharge outlet of each tank. Analysis was done to obtain the pH, temperature, and concentrations of BOD_5_, COD_K2Or2O7_, SS_total_, oil and grease, and true color to evaluate the treatment efficiency of the proposed system.

For the SEM observation, the samples were treated as follows: (1) fixation by Glutardialdehyde solution (2.5%) under 4 °C for 2.5 h; (2) flushing by phosphoric acid buffer solution (0.1 M) three times for 10 min, and (3) dewatering eight times with different concentrations of ethyl alcohol solutions (30%, 50%, 70%, 90%, 95%, 100%, 100% and 100%) for 10 min. Microbe growth in the AS tank and CA tank was then observed and recorded using SEM.

### 2.4. Denatured Gradient Gel Electrophoresis (DGGE)

The DGGE was applied to analyze the biological phase. Microbes were extracted from the sludge samples taken from each tank, after which various chemicals (lysis buffer, lysozyme, proteinase, sodium dodecyl sulfate, NaCl, cetyl trimethyl ammonium bromide, chloroform/Isoamyl alcohol, isopropanol, ethanol) were added to chemically break down the cell walls. A physically quick-freezing procedure was applied, followed by rapid shaking in a warm bath (with 37 °C for 1 h) to destroy the cell wall. The sample was then placed in a centrifuge (12,000 rpm for 15 min) to separate the proteins, polysaccharides, and DNA material. Then, several other chemicals (10X PCR buffer, dNTP mixture, forward primer, reverse primer, taq polymerase, template, autoclaced dH2O) were applied to stimulate PCR and amplify the amount of DNA material. Finally, electrophoresis (with 160 V, 60 °C for 180 min) and silver staining (for 20 min) were used to complete the DGGE experiment. For the biological content analysis, the sludge recycling ratio of 0.3 was maintained. The changes in the amount of suspended and fixed microorganisms in the AS tank and CA tank were monitored.

## 3. Results and Discussion

### 3.1. Patterns of Temperature, pH, and CDO in the AS/CA System

Under the sludge-growing conditions in the cultivation period, the sludge-returning pump returned almost all of the sludge because the quantity of sludge that flowed from the aeration tank to the sedimentation tank was relatively small during the cultivation period. However, the quantity of sludge in the front AS system exceeded that in the rear AS/CA system, indicating that the AS system without filters generates more sludge which flows to the sedimentation tank more easily. Under the sludge-growing conditions in the stable period, the sedimentation of the sludge in the sedimentation tank exhibited excellent settlement of the sludge, and the system was very stable. The sludge at the rear AS/CA tank was darker than that in the other tank, revealing that the AS/CA tank contained peeled-off anaerobic biofilms.

Under the sludge-growing conditions with 120 mg/L of oil and grease added, both sedimentation tanks contained black sludge, indicating that the sludge became covered by the oil and grease and could not breathe, entering the anaerobic state; some of the sludge stuck to the wall of the sedimentation tank. However, the sludge in the rear AS/CA system was black and yellow, indicating that the AS/CA system can resist the oil and grease more effectively than can the AS system, and it provides better biological growing conditions. In Figure 2, the times required by the AS system and the AS/CA system to cultivate microorganisms stably are almost the same, but the COD removal rate of the AS/CA system exceeded that of the AS system by 2–7% in the stable period. Increasing the oil and grease load from 15 mg/L to 120 mg/L increases the difference between the COD removal rates of the two systems to 20%, potentially explaining why the AS/CA system contained more microorganisms and more species. When the tests were conducted after the addition of oil and grease in a later stage, the outflow COD value was increased, revealing that oil and grease affected the growth of microorganisms, which have a maximum tolerance for the Wei Lih fragrance oil of about 120 mg/L. The AS/CA system obviously clearly exhibits greater tolerance for oil and grease tolerance than does the AS system.

Figure 3 shows the pH profile of the developed AS system, where the pH of the raw water dropped to a slightly acidic level, revealing that the production of organic acid and the nitrification of intermediate products such as organic acids, nitrates, or nitrites extended the operating time of the system and promoted biodegradation, tending to increase the pH to the neutral level.

Figure 4 shows the pH profile of the developed AS/CA system used in this study. The pH of the raw water dropped to a slightly acidic level, revealing that the production of organic acid and the nitrification of intermediate products such as organic acids, nitrates, or nitrites extended the operating time of the system and promoted biodegradation, tending to increase the pH to the neutral level.

Figure 5 shows the temperature profile of the AS system used in this study. The temperature dropped during the reaction process, indicating that both systems exhibited greater bio-degeneration and generated more heat (at an earlier stage OR earlier in the process). Both systems were established under a fume hood and so heat was effectively dissipated. Water flowed for longer later in the process, so the heat dissipation time was longer and the water temperature was lower.

Figure 6 shows the temperature profiles of the developed AS/CA system used in this study. The temperature dropped during the reaction process, indicating that both systems exhibited greater bio-degeneration and generated more heat (at an earlier stage OR earlier in the process). Both systems were established under a fume hood and so heat was effectively dissipated. Water flowed for longer later in the process, so the heat dissipation time was longer and the water temperature was lower.

### 3.2. Removal of BOD, COD, SS, and True Color by the AS/CA System

Table 6 summarizes the analysis of BOD in the stable period. Regardless of whether the AS system or the AS/CA system was used, the BOD in the synthetic domestic sewage decomposed completely. The AS/CA system was slightly more effective and the difference between the BOD removal rates of the two systems was 1%–2%.

Table 7 concerns the analysis of SS in the stable period. Regardless of whether the AS system or the AS/CA system was used, the SS in the synthetic domestic sewage decomposed completely. The AS/CA system was slightly more effective, and the SS-out was lower than AS system.

In Table 8, both the AS and the AS/CA systems exhibit true color removal. The AS/CA system was slightly more effective, and the difference between the color removal rates of the two systems was 3–8%. The transparency and clarity of the liquid at the top of the sedimentation tank in the AS/CA system was higher than in the AS system.

Since the BOD averages of the inflowing and outflowing water in the AS system were 213 mg/L and 5 mg/L, respectively, the growth coefficient was Y = ΔX/ΔS ≒ 0.5, the chemical formula of the aerobic microorganisms was C_5_H_7_O_2_N, and the concentration N in the water that was consumed in the growth process of the microorganisms is calculated to be 12.8 mg/L. In Table 9, the total concentration of org.-N, NH4^+^-N, NO2^−^-N, and NO3^−^-N in the inflow water is slightly higher than the total concentration of org.-N, NH4^+^-N, NO2^−^-N, and NO3^−^-N in the outflow water, and 3.37 mg/L of N in the water was consumed during the growth of microorganisms. This biological treatment system exhibits almost no de-nitration effect, and in the overall AS system, any trace of N is eliminated, perhaps because of the de-nitration effect of some of the anaerobic sludge in the sedimentation tank.

Since the BOD averages of the inflow and outflow water of the AS/CA system are 213 mg/L and 3 mg/L, respectively, the growth coefficient was Y = ΔX/ΔS ≒ 0.25, the chemical formula of the aerobic microorganisms was C_5_H_7_O_2_N, and the concentration N in the water that is consumed in the growth process of the microorganisms is calculated to be 6.50 mg/L. In Table 5, the total concentration of org.-N, NH4^+^-N, NO2^−^-N, and NO3^−^-N in the inflow water is higher than that of org.-N, NH4^+^-N, NO2^−^-N, and NO3^−^-N in the outflow water, and 10.04 mg/L of N in the water is consumed during the growth of microorganisms, so this system exhibits the de-nitration effect.

As shown in Figure 7, both the AS system and the AS/CA system decompose the Wei Lih fragrance oil with an inflow concentration of 30 mg/L or less, and the concentration of the outflowing oil and grease is maintained below 10 mg/L. When the inflow concentration is increased to 60 mg/L, the concentration of the outflowing oil and from the AS system exceeds the discharge standard maximum of 10 mg/L, but the concentration of the outflowing oil and grease from the AS/CA system remains below standard maximum. When the inflow concentration is further increased to 120 mg/L, the outflow concentrations of both systems exceed the discharge maximum, but both systems remove some oil and grease. The AS/CA system is observed to be more effective than the AS system in this respect; the removal ratio of the AS/CA system is 6–11% higher than that of the AS system. The blackening of the sludge in the sedimentation tank in the AS/CA system was not as severe as that in the AS system, revealing that the AS/CA system exhibits a higher tolerance of highly concentrated oil and grease.

As shown in Table 10, the AS system has a higher microorganism load F/M value and a smaller volume load than the AS/CA system. COD loading of the AS system is 0.68 kg·COD/Kg·MLSS·day higher the AS/CA system 0.58 kg·COD/kg·MLSS·day. BOD loading of the AS system is 0.36 kg·BOD/kg·MLSS·day higher the AS/CA system 0.31 kg·BOD/kg·MLSS·day. COD volume loading of the AS system is 1.17 kg·COD/m^3^·day smaller than the AS/CA system 1.19 kg·COD/m^3^·day. BOD volume loading of the AS system is 0.62 kg·BOD/m^3^·day smaller than the AS/CA system 0.63 kg·BOD/m^3^·day.

### 3.3. SEM Photographs of the Biological Phases

From the SEM photographs of the biological phase of the first and second tanks of the AS system in the stable period and the distribution of the suspended and fixed biological phases of the first and second tanks in the AS/CA system in the stable period (as shown in Figure 8), the AS/CA system supports a higher bacteria growth density than does the AS system, and the microbial groups differ: The round-headed segmented bacterium in the first tank was longer than the suspended bacteria in the second tank, and the fixed bacterium in the second tank was flat-headed and non-segmented. These observations demonstrate that the AS/CA system contains more diverse microbes, possibly explaining its higher treatment efficiency. From the SEM photographs of the biological phases of the AS and AS/CA systems after the addition of 120 mg/L oil and grease [8], displayed in Figure 9, the surfaces of both systems were covered by oil films, but the AS system was primarily covered by bacilli and cocci, whereas the AS/CA system was primarily covered by round-headed and flat-headed bacteria of various lengths.

### 3.4. DGGE in AS System and AS/CA System

Figure 10 compares the DGGE of the AS system with that of the AS/CA system. The numerals (3, 4, 5 and 8, 9, 10) indicate the suspended and fixed microorganisms in the first and second tanks of the AS/CA system, respectively. The number of transverse stripes clearly exceeds that indicated by the numbers (1, 2) and (6, 7) in the AS system, revealing that the species of microorganisms were more numerous. When 120 mg/L of oil and grease were added, the number of the transverse stripes in each reaction tank increased slightly, indicating that the size of the microbial group in each reaction tank increased. This increase in the AS/CA system exceeded that in the AS system.

## 4. Conclusions

The AS/CA system which contained both suspended and attached microorganisms was characterized by multiple biological phases, and longer food chains of microorganisms was thus a stable system with powerful treatment function [9]. When the inflow concentration of oil and grease was increased to 120 mg/L, the results of the COD experiment reveal that the AS/CA system is affected less than the AS system, and the COD removal rate of the AS/CA system is maintained at 81%, which exceeds that (61.5%) of the AS-only system. Therefore, the AS/CA system tolerates oil and grease better than the AS system. When the influent oil and grease concentration reached 60 mg/L, the effluent concentration of the AS system was 13 mg/L, which exceed 10 mg/L, the Taiwan Effluent Standard for oil and grease. However, in the AS/CA system, the effluent oil and grease concentration was 8 mg/L, which was below the required standard. Both the AS and the AS/CA systems exhibit true color removal. The AS/CA system is slightly more effective, and the difference between the color removal rates of the two systems is 3%–8%. The transparency and clarity of the liquid at the top of the sedimentation tank in the AS/CA system was higher than in the AS system.

## Figures and Tables

**Figure 1 ijerph-17-00544-f001:**
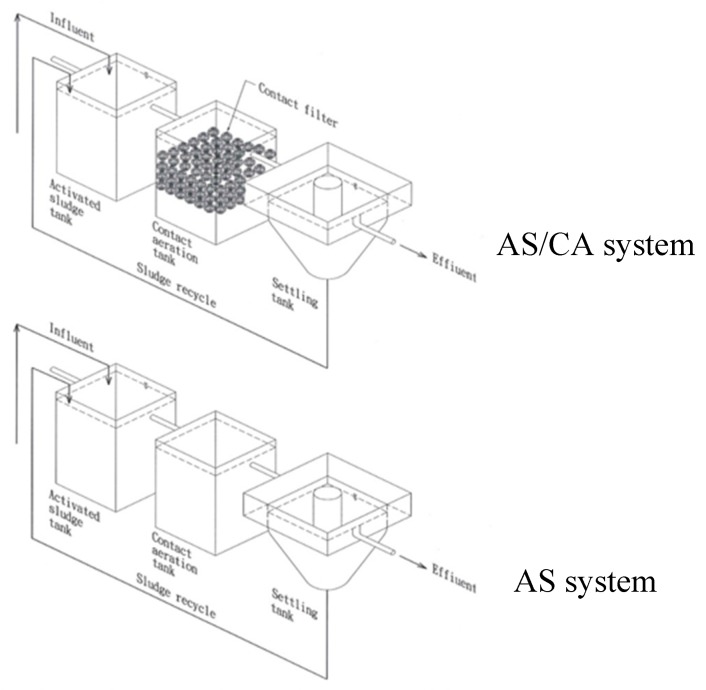
Flow chart of small model factory with activated sludge (AS) and activated sludge/contact aeration (AS/CA) systems.

**Figure 2 ijerph-17-00544-f002:**
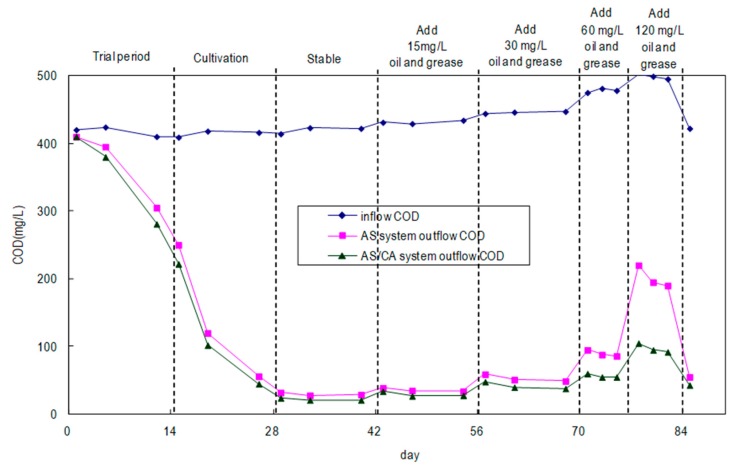
COD of synthetic domestic sewage treated by AS and AS/CA systems.

**Figure 3 ijerph-17-00544-f003:**
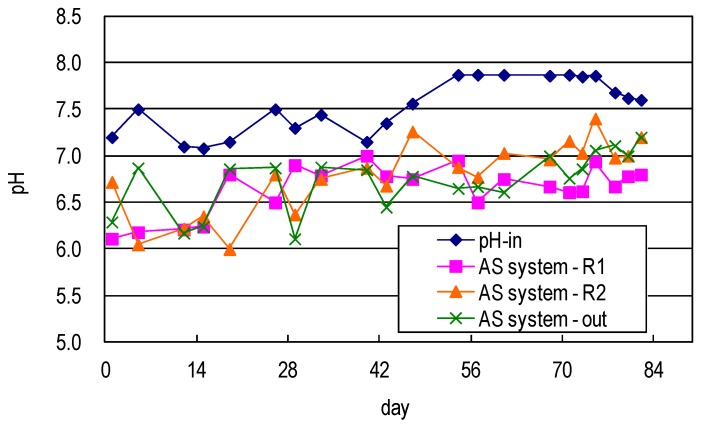
pH value of synthetic domestic sewage treated using AS system.

**Figure 4 ijerph-17-00544-f004:**
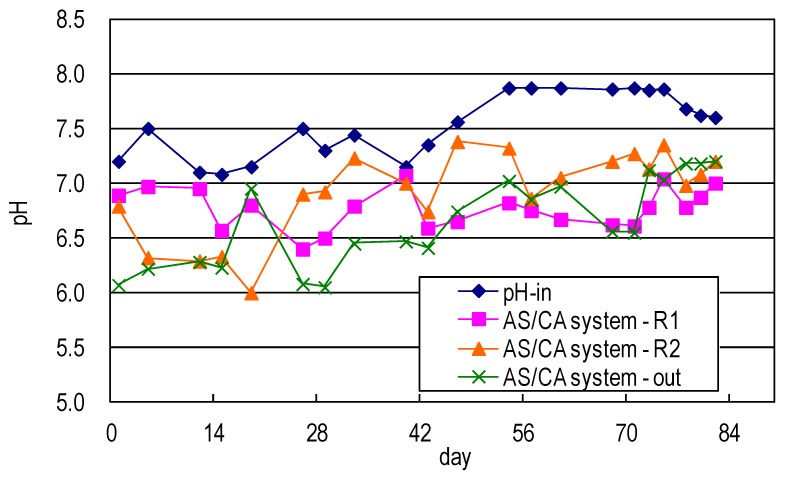
pH value of synthetic domestic sewage treated using AS/CA system.

**Figure 5 ijerph-17-00544-f005:**
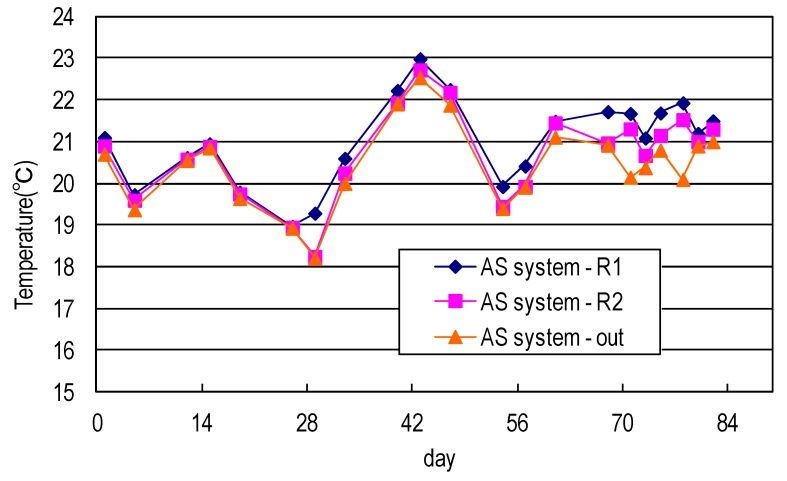
Temperature of synthetic domestic sewage treated using AS system.

**Figure 6 ijerph-17-00544-f006:**
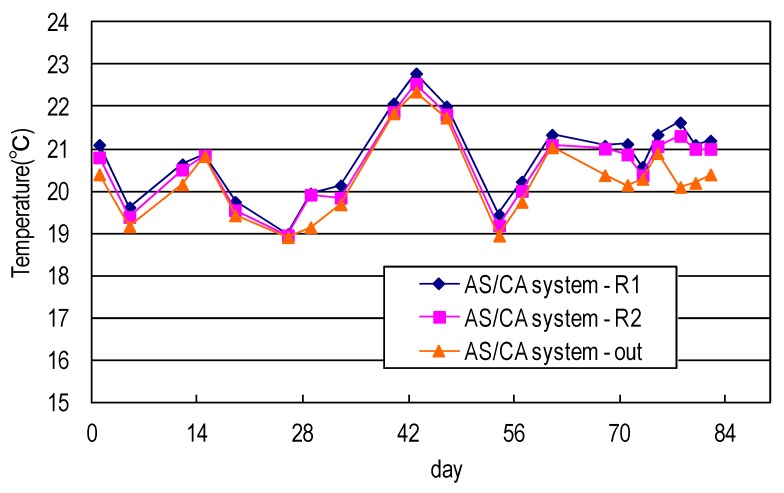
Temperature of synthetic domestic sewage treated using AS/CA system.

**Figure 7 ijerph-17-00544-f007:**
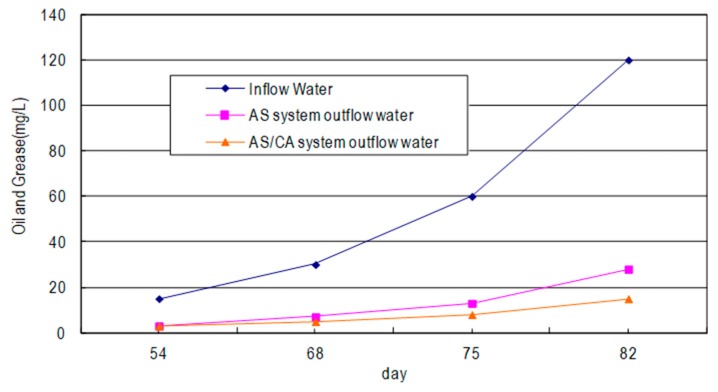
Treating oil and grease using AS system and AS/CA system.

**Figure 8 ijerph-17-00544-f008:**
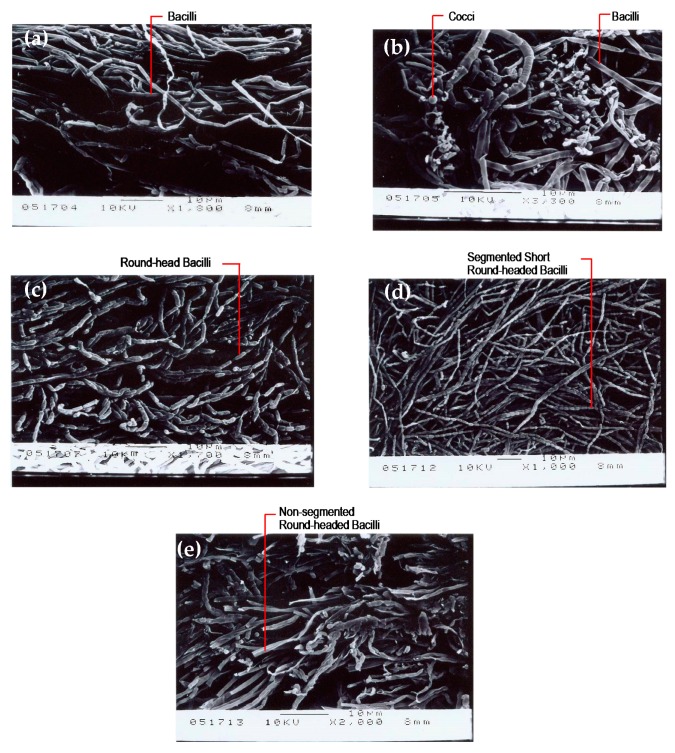
(**a**) Biological phase of first tank in AS system during stable period; (**b**) biological phase of second tank in AS system during stable period; (**c**) biological phase of first tank in AS/CA system during stable period; (**d**) suspended biological phase of second tank in AS/CA system during stable period; (**e**) fixed biological phase of second tank in AS/CA system during stable phase.

**Figure 9 ijerph-17-00544-f009:**
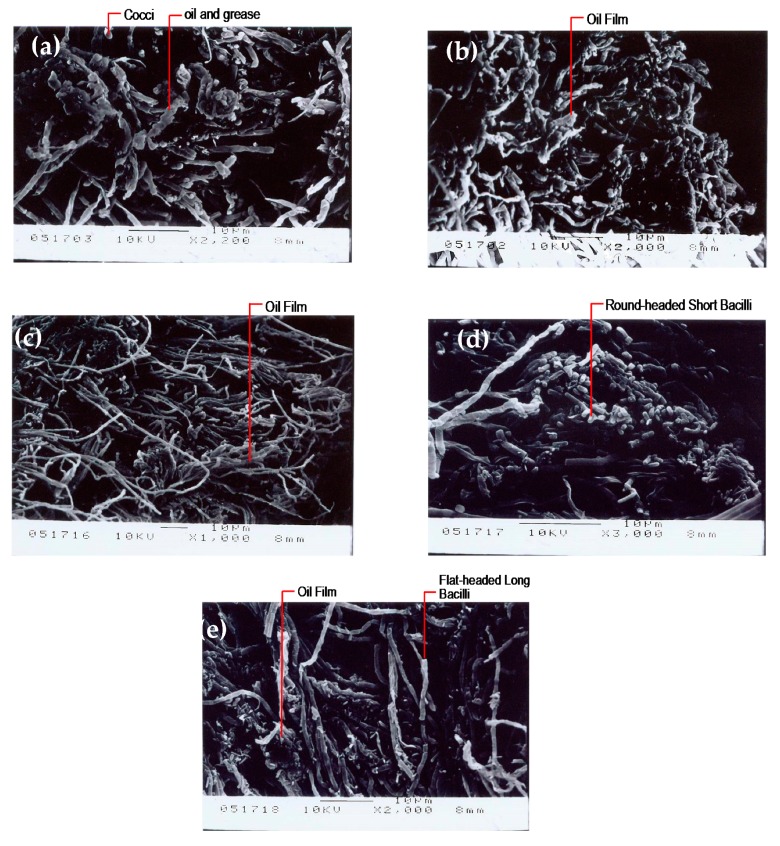
(**a**) Biological phase of first tank in AS system following addition of 120 mg/L of oil and grease; (**b**) biological phase of second tank in AS system following addition of 120 mg/L of oil and grease; (**c**) biological phase of first tank in AS/CA system following addition of 120 mg/L of oil and grease; (**d**) suspended biological phase of second tank in AS/CA system following addition of 120 mg/L of oil and grease; (**e**) fixed biological phase of second tank in AS/CA system following addition of 120 mg/L of oil and grease.

**Figure 10 ijerph-17-00544-f010:**
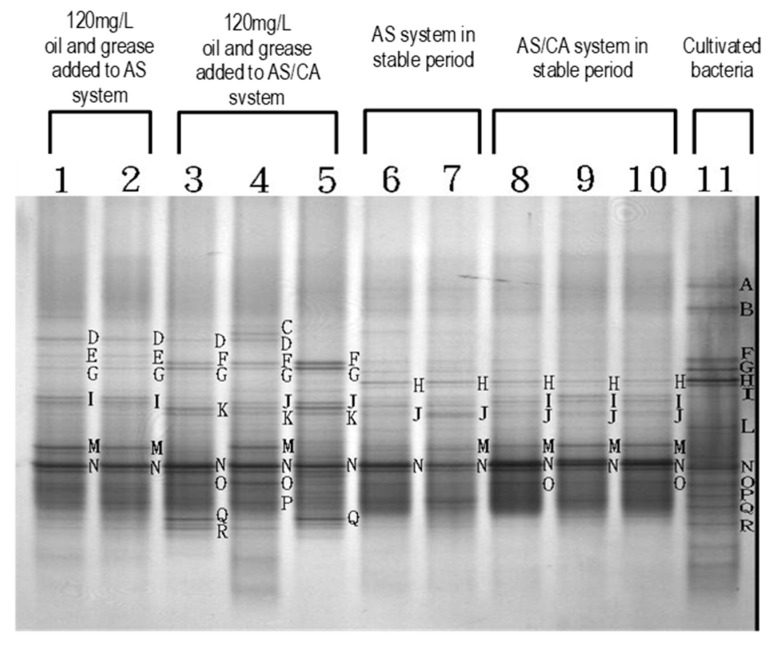
Denatured gradient gel electrophoresis (DGGE) in AS system and AS/CA system after 120 mg/L of oil and grease are added and in stable period.

**Table 1 ijerph-17-00544-t001:** Composition of synthetic domestic sewage.

Sewage Composition	Concentration (mg/L)
Glucose	400
Nutritious Beef Juice	49
Sodium Carbonate	(Depend on pH value)
Iron Chloride	12
Ammonium Chloride	38
Dipotassium Hydrogen Phosphate	14
Sodium Sulfite	17
Average COD	420
Average BOD	213

**Table 2 ijerph-17-00544-t002:** Chemical compositions of Wei Lih fragrance oil used in this study (inflow parameters values are 15 mg/L, 30 mg/L, 60 mg/L, and 120 mg/L).

Composition	Percentage (%)
Meat Coronene Acid	1.40–2.00
Palmitic Acid	18.05–20.98
Palmitoleic Acid	3.01–3.95
Stearic Acid	3.15–5.91
Oleic Acid	48.76–55.33
Linoleic Acid	13.20–18.31
α-Linoleic Acid	Trace

**Table 3 ijerph-17-00544-t003:** Composition and properties of emulsifier used in small model factory (inflow parameters values are 1.5 mg/L, 3.0 mg/L, 6.0 mg/L, and 12.0 mg/L).

1. Brand and Product Number: Sigma-Aldrich 274348
2. Chemical Name: Polyethylene glycol sorbitan monolaurate
3. Average Mol. Wt.: 1228
4. Composition: Lauric acid, 50% (with balance primarily myristic, palmitic, and setaric acids)
5. Fp > 230 °F
6. Density: 1.095 g/mL at 25 °C

**Table 4 ijerph-17-00544-t004:** Specifications of biological contact filter in model factory (filling rate of biological contact filter in CA tank is 50%).

Model	Material	Size D × H (mm)	Density (kg/m^3^)	Specific Surface Area (m^2^/m^3^)	Porosity (%)	Number of per Unit Volume (pieces/m^3^)
Double Star ball	PVC	32 × 27	76	274	91	74,000

Note: PVC: Polyvinyl Chloride.

**Table 5 ijerph-17-00544-t005:** Size and effective volume of each tank in small model factory.

Item	Dimensions L × W × H (cm)	Effective Volume (Liter)	No. of Tanks
Activated Sludge Tank	15 × 15 × 20	4.50	2
Contact Aeration Tank	15 × 15 × 20	4.50	2
Sediment Tank Square Upper Half	15 × 15 × 10	2.25	2
Conical Lower Half	15 × 15 × 8	0.58	2

**Table 6 ijerph-17-00544-t006:** BOD of synthetic domestic sewage treated using AS system and AS/CA system.

Period	No. of Days	BOD-In (mg/L)	AS System		AS/CA System	
			BOD-Out (mg/L)	Removal Rate (%)	BOD-Out (mg/L)	Removal Rate (%)
Stable period	29	212	5	97.64	3	98.58
	33	203	3	98.52	2	99.01
(14 days)	40	224	7	96.88	3	98.66
Mean Average		213	5	97.65	3	98.59

**Table 7 ijerph-17-00544-t007:** Quantity of sludge and SS in synthetic domestic sewage treated using AS system and AS/CA System.

Period	No. of Days	AS System			AS/CA System		
		SS-R1 (mg/L)	SS-R2 (mg/L)	SS-Out (mg/L)	SS-R1 (mg/L)	SS-R2 (mg/L)	SS-Out (mg/L)
Stable period	29	1790	1670	18	1720	2260	12
	33	1760	1560	15	1680	2380	9
(14 days)	40	1820	1740	12	1750	2540	7
Mean Average		1790	1657	15	1717	2393	9

**Table 8 ijerph-17-00544-t008:** True color of synthetic domestic sewage treated using AS system and AS/CA system.

Period	No. of Days	True Color-In(Unit)	AS System		AS/CA System	
			True Color-Out(Unit)	Removal Rate (%)	True Color-Out (Unit)	Removal Rate (%)
Stable period	29	125	35	72	30	76
	33	118	27	77	25	79
(14 days)	40	130	40	69	30	77
Mean average		124	34	73	28	77

**Table 9 ijerph-17-00544-t009:** Nitration of synthetic domestic sewage treated using AS system and AS/CA system in stable Stage.

Period	No. of Days	Inflow Water				AS System				AS/CA System			
		Org.-N	NH_4_^+^-N	NO_3_^−^-N	NO_2_^−^-N	Org.-N	NH_4_^+^-N	NO_3_^−^-N	NO_2_^−^-N	Org.-N	NH_4_^+^-N	NO_3_^−^-N	NO_2_^−^-N
Stable period	29	7.89	9.3	1.1	0.03	1.21	0.37	0.05	0.02	2.28	0.06	0.02	0.02
	33	7.31	9.9	1.08	0.02	1.65	0.22	0.04	0.01	1.01	0.02	0.01	0.02
(14 days)	40	8.1	8.9	1.06	0.02	1.92	0.64	0.05	0.02	1.64	0.02	0.01	0.01
Average		7.77	9.37	1.08	0.02	1.59	0.41	0.05	0.02	1.64	0.03	0.01	0.02

**Table 10 ijerph-17-00544-t010:** Comparison of performance of treatment of synthetic domestic sewage using AS system and AS/CA system.

Item	Inflow	AS System		AS/CA System	
		First Tank	Second Tank	First Tank	Second Tank
SS (mg/L)		1790	1657	1717	2393
		1724		2055	
COD (mg/L)	420	30		22	
BOD (mg/L)	213	5		3	
kgCOD/kg.MLSS.day		0.68		0.58	
kgBOD/kg.MLSS.day		0.36		0.31	
kgCOD/m^3^.day		1.17		1.19	
kgBOD/m^3^.day		0.62		0.63	

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
