# Peer review of "Feasibility Study of Activated Sludge/Contact Aeration Combined System Treating Oil-Containing Domestic Sewage"

_ijerph, 2020, doi:10.3390/ijerph17020544_

Round 1

Reviewer 1 Report

This manuscript investigated the feasibility of activated sludge/contact aeration combined system treating oil-containing domestic sewage.  However, there are many problems in this manuscript.  They are noted here for the authors’ references:

The Introduction doesn’t provide sufficient background and include all relevant references. It must be improved. The full names of the abbreviations should only be provided once when they are presented for the first time. Materials and Methods must be improved. Many important information, like the used biological contact filters, the oil, grease, the emulsifier, influent composition and quality, and the analyzing methods, should be provided in detail.   The number of significant digits for the effective volume should be unified. AS and AS/CA should be noted in Figure 1. Line 59-66: The running parameters cannot be sub sections. Table 2 – Table 6: All the parameters of Line Inflow should be provided. The analyzing tools for the microorganisms are traditional, and the results analyses are unpersuasive.

Author Response

Itemized Replies to Reviewer’s Comments

Re: manuscript “Feasibility study of activated sludge/contact aeration combined system treating oil-containing domestic sewage” by Chen et al. (Manuscript ID: ijerph-638857)

Thank you very much. We appreciated for your comments.

Revise 1

The Introduction doesn’t provide sufficient background and include all relevant references.

A: line 45.

This is to enhance the removal of COD, oil and grease and true color and would require a smaller land area.

The full names of the abbreviations should only be provided once when they are presented for the first time.

A: Deleted line 45 and line 77, combined activated sludge/contact aeration. →AS/CA

Materials and Methods must be improved. Many important information, like the used biological contact filters, the oil, grease, the emulsifier, influent composition and quality

A: p.2,

Table 1. Composition of Synthetic Domestic Sewage

Sewage Composition

Concentration (mg/L)

Glucose

400

Nutritious Beef Juice

49

Sodium Carbonate

(Depend on pH value)

Iron Chloride

400

Ammonium Chloride

49

Dipotassium Hydrogen Phosphate

12

Sodium Sulfite

38

Average COD

14

Average BOD

17

Table 2. Composition of Wei Lih Fragrance Oil

Composition

Percentage (%)

Meat Coronene Acid

1.40~2.00

Palmitic Acid

18.05~20.98

Palmitoleic Acid

3.01~3.95

Stearic Acid

3.15~5.91

Oleic Acid

48.76~55.33

Linoleic Acid

13.20~18.31

α-Linoleic Acid

Trace

Table 3. Composition and Properties of Emulsifier

1. Brand and Product Number: Sigma-Aldrich 274348

2. Chemical Name: Polyethylene glycol sorbitan monolaurate

3. Average Mol. Wt.: 1,228

4. Composition: Lauric acid, 50%

(with balance primarily myristic, palmitic, and setaric acids)

5. Fp > 230℉

6. Density: 1.095 g/mL at 25℃

Table 4. Specifications of Biological Contact Filter in Model Factory

Model

Material

Size

D×H

(mm)

Density

(kg/m3)

Specific Surface Area

(m2/m3)

Porosity

(%)

Number of per Unit Volume

(pieces/m3)

Double

Star ball

PVC

32×27

76

274

91

74,000

the analyzing methods

A: The analyze water quality based on “the standard methods for the examination of water and wastewater.”

AS and AS/CA should be noted in Figure 1.

A: p.3, Figure 1.

AS system

AS/CA system

Line 66-76: The running parameters cannot be sub sections.

A: p.3,

AS system

AS/CA system

Controlled Conditions of Small Model Factory:

2.1 Inflow Rate: 18.75 C.C./min (as for Biological Aeration Tank (Hydraulic Retention Time (HRT): 8 hrs)

2.1.1 HRT of Activated Sludge Tank: 4 hrs

2.1.2 HRT of Contact Aeration Tank: 4 hrs

2.1.3 HRT of Sedimentation Tank: 2.5 hrs

2.2 Sludge Return Rate: 30 %

2.3 Sludge Retention Time (SRT)≒15 days

2.4 Filling Rate of Biological Contact Filter in CA Tank: 50%.

Figure 1. Flow Chart of Small Model Factory with AS and AS/CA Systems.

Table 2 – Table 6: All the parameters of Line Inflow should be provided.

A: The inflow rate is 18.75 c.c./min.

Table 1. Composition of Synthetic Domestic Sewage

Sewage Composition

Concentration (mg/L)

Glucose

400

Nutritious Beef Juice

49

Sodium Carbonate

(Depend on pH value)

Iron Chloride

400

Ammonium Chloride

49

Dipotassium Hydrogen Phosphate

12

Sodium Sulfite

38

Average COD

14

Average BOD

17

Table 2. Composition of Wei Lih Fragrance Oil

Composition

Percentage (%)

Meat Coronene Acid

1.40~2.00

Palmitic Acid

18.05~20.98

Palmitoleic Acid

3.01~3.95

Stearic Acid

3.15~5.91

Oleic Acid

48.76~55.33

Linoleic Acid

13.20~18.31

α-Linoleic Acid

Trace

Table 3. Composition and Properties of Emulsifier

1. Brand and Product Number: Sigma-Aldrich 274348

2. Chemical Name: Polyethylene glycol sorbitan monolaurate

3. Average Mol. Wt.: 1,228

4. Composition: Lauric acid, 50%

(with balance primarily myristic, palmitic, and setaric acids)

5. Fp > 230℉

6. Density: 1.095 g/mL at 25℃

Reviewer 2 Report

Review for “Feasibility study of activated sludge/contact aeration combined system treating oil-containing domestic sewage”.

The manuscript by Chen et al. attempts to compare the treatment efficiency of oil-containing domestic sewage by AS or AS/CA systems, and found that the combined AS/CA system has much higher efficiency than AS. Overall, this is a solid study with interesting observation. However, before acceptance of this manuscript, I suggest the authors to address my following concerns.

Major:

Introduction section is not thorough, and does not contain enough state-of-the-art literatures. The methods part does not contain all details, for example, DGGE method? BOD measurement method? Etc. Importantly, I suggest the authors to test bacteria diversity such as based on 16S-rDNA analysis. The authors mainly presented information about bacteria by SEM, which cannot disclose enough information about bacteria species.

Minor:

The authors need to pay attention to the language, especially grammar errors.

Author Response

Itemized Replies to Reviewer’s Comments

Re: manuscript “Feasibility study of activated sludge/contact aeration combined system treating oil-containing domestic sewage” by Chen et al. (Manuscript ID: ijerph-638857)

Thank you very much. We appreciated for your comments.

Revise 2

Introduction section is not thorough, and does not contain enough state-of-the-art literatures.

A: line 45, This is to enhance the removal of COD, oil and grease and true color and would require a smaller land area.

DGGE method?

A: p.3, line 89,

DGGE analysis: DGGE was used to analyze the biological phase. DNAs were extracted from the sludge samples taken from each tank, after which various chemicals were added to breakdown chemically the cell walls. Using physically quick freezing method procedure, rapid shaking bath warm up to destroy the cell wall, and put the sample into a centrifuge to separate the proteins, polysaccharide, and DNA material. Then using several other chemicals to stimulate PCR,and amplify the amount of DNA material. Finally electrophoresis and silver staining were used to complete the DGGE experiment.

BOD measurement method?

A: BOD measurement method based on “the standard methods for the examination of water and wastewater”.

I suggest the authors to test bacteria diversity such as based on 16S-rDNA analysis. The authors mainly presented information about bacteria by SEM, which cannot disclose enough information about bacteria species.

A: DGGE can disclose enough information about bacteria species.

Reviewer 3 Report

The article is interesting and includes the novelty aspect. After making minor changes meet the publication conditions.

Remarks:

The abstract should include the explicit aim of research and should be supplemented with specific values of obtained test results.

The review of the literature should be broadened taking into account the results of research obtained by other scientists and relating to the research conditions carried out by the authors.

In the case of the research methodology, please extend the point to include in Table 1 and Figure 1 together with an explanation of the justification for placing them in the text.

Author Response

Itemized Replies to Reviewer’s Comments

Re: manuscript “Feasibility study of activated sludge/contact aeration combined system treating oil-containing domestic sewage” by Chen et al. (Manuscript ID: ijerph-638857)

Thank you very much. We appreciated for your comments.

Revise 3

The abstract should include the explicit aim of research.

A: The aim of research is Title: Feasibility study of activated sludge/contact aeration combined system treating oil-containing domestic sewage.

The abstract should be supplemented with specific values of obtained test results.

A: p.1, line 13, When the influent oil and grease concentration reached 60 mg/L, the effluent concentration of the AS system was 13 mg/L, which exceed 10 mg/L, the Taiwan Effluent Standard for oil and grease. However, in the AS/CA system, the effluent oil and grease concentration was 8 mg/L, which was below the required standard. Therefore, the AS/CA system tolerates oil and grease better than does the AS system.

Please extend the point to include in Table 1 and Figure 1 together with an explanation of the justification for placing them in the text.

A: Table 1 displays the size and effective volume of each water tank. It can check HRT of Activated Sludge Tank, contact Aeration Tank and Sedimentation Tank.

Figure 1. shows Flow Chart of Small Model Factory with AS and AS/CA systems.

Reviewer 4 Report

Reviewer’s comments

The manuscript no. ijerph-638857 describes a AS/CA system, which has been used to treat oil-containing domestic sewage. However, the paper is poorly written with several minor mistakes. I suggest this study is rewritten and carefully concluded to show findings.

Background of this study is not clear. Material and methods part in not well solved. What kind of analytical methods did you use for this study? Did you analyze water quality based on "the standard methods for the examination of water and wastewater"? Wastewater treatment is affected by water temperature and pH. Any correlations between temperature/pH and water quality parameters? What is the material of biological contact filter in CA tank? Retained sludge concentration in SS-R2 between AS system (1500-1700 mg/L) and SS-R2 (2200-2500mg/L) is much difference. It is not quite fair to compare the both results. The descriptions are confusing and sometimes repeated. Moreover, there are some language errors in the text. Please ask a native to get through the manuscript.

Author Response

Itemized Replies to Reviewer’s Comments

Re: manuscript “Feasibility study of activated sludge/contact aeration combined system treating oil-containing domestic sewage” by Chen et al. (Manuscript ID: ijerph-638857)

Thank you very much. We appreciated for your comments.

Revise 4

Material and methods part in not well solved.

A: p.2,

Table 1. Composition of Synthetic Domestic Sewage

Sewage Composition

Concentration (mg/L)

Glucose

400

Nutritious Beef Juice

49

Sodium Carbonate

(Depend on pH value)

Iron Chloride

400

Ammonium Chloride

49

Dipotassium Hydrogen Phosphate

12

Sodium Sulfite

38

Average COD

14

Average BOD

17

Table 2. Composition of Wei Lih Fragrance Oil

Composition

Percentage (%)

Meat Coronene Acid

1.40~2.00

Palmitic Acid

18.05~20.98

Palmitoleic Acid

3.01~3.95

Stearic Acid

3.15~5.91

Oleic Acid

48.76~55.33

Linoleic Acid

13.20~18.31

α-Linoleic Acid

Trace

Table 3. Composition and Properties of Emulsifier

1. Brand and Product Number: Sigma-Aldrich 274348

2. Chemical Name: Polyethylene glycol sorbitan monolaurate

3. Average Mol. Wt.: 1,228

4. Composition: Lauric acid, 50%

(with balance primarily myristic, palmitic, and setaric acids)

5. Fp > 230℉

6. Density: 1.095 g/mL at 25℃

Table 4. Specifications of Biological Contact Filter in Model Factory

Model

Material

Size

D×H

(mm)

Density

(kg/m3)

Specific Surface Area

(m2/m3)

Porosity

(%)

Number of per Unit Volume

(pieces/m3)

Double

Star ball

PVC

32×27

76

274

91

74,000

p.3, line 89,

DGGE analysis: DGGE was used to analyze the biological phase. DNAs were extracted from the sludge samples taken from each tank, after which various chemicals were added to breakdown chemically the cell walls. Using physically quick freezing method procedure, rapid shaking bath warm up to destroy the cell wall, and put the sample into a centrifuge to separate the proteins, polysaccharide, and DNA material. Then using several other chemicals to stimulate PCR,and amplify the amount of DNA material. Finally electrophoresis and silver staining were used to complete the DGGE experiment.

What kind of analytical methods did you use for this study? Did you analyze water quality based on "the standard methods for the examination of water and wastewater"?

A: The analytical methods have COD, BoD, SS, Ture Color, Org-N, NH4+-N, NO3--N, NO2--N, Oil and Grease, pH and temperature. They all based on “the standard methods for the examination of water and wastewater”.

Wastewater treatment is affected by water temperature and pH. Any correlations between temperature/pH and water quality parameters?

A: p.5,

As presented in Figure 3 and 4, regardless of whether the AS system or the AS/CA system is utilized, the pH of the raw water drops to a slightly acidic level, revealing that the production of organic acid and the nitrification of intermediate products such as organic acids, nitrates or nitrites extends the operating time of the system and promotes biodegradation, tending to increase the pH to the neutral level.

Figure 3. pH Value of Synthetic Domestic Sewage Treated Using AS System

Figure 4. pH Value of Synthetic Domestic Sewage Treated Using AS/CA System

In Figure 5 and 6, whether the AS system or the AS/CA system is used, the temperature drops during the reaction process, indicating that both systems exhibited greater bio-degeneration and generate more heat (at an earlier stage OR earlier in the process). Both systems were established under a fume hood and so heat was effectively dissipated. Water flowed for longer later in the process, so the heat dissipation time was longer, and the water temperature was lower.

Figure 5. Temperature of Synthetic Domestic Sewage Treated Using AS System 

Figure 6. Temperature of Synthetic Domestic Sewage Treated Using AS/CA System

What is the material of biological contact filter in CA tank?

A: p.3,

Table 4. Specifications of Biological Contact Filter in Model Factory

Model

Material

Size

D×H

(mm)

Density

(kg/m3)

Specific Surface Area

(m2/m3)

Porosity

(%)

Number of per Unit Volume

(pieces/m3)

Double

Star ball

PVC

32×27

76

274

91

74,000

Retained sludge concentration in SS-R2 between AS system (1500-1700 mg/L) and SS-R2 (2200-2500mg/L) is much difference. It is not quite fair to compare the both results. The descriptions are confusing.

A: The Influent, HRT and Sludge Return Rate of AS and AS/CA system are the same. Sludge concentration of AS/CA system is higher than AS system. Therefore, the AS/CA system tolerates oil and grease better than does the AS system.

There are some language errors in the text. Please ask a native to get through the manuscript.

A: We ask a native to get through the manuscript. Best English Editing Service, BEES. Efan graduated cum luade from the University of Masschusetts at Amherst with a BS in Environmental Science and Biology.

Round 2

Reviewer 1 Report

Although the authors did some revisions, several comments aren’t well responded.  They are noted here for the authors’ references once more:

The writing of Introduction is both abnormal and too simple. It doesn’t provide sufficient background introduction of the related studies, and doesn’t clearly note the meaning of this study. The number of significant digits for the effective volume in Table 5 and the texts should be unified. Line 69-76: The running parameters cannot become sub-Sections. They could be changed to normal texts. Table 2 – Table 6: Each inflow parameters’ values for all the lines should be provided in the tables. The appearances of the tables and the figures are ugly. Their format should be uniformed according to the journal requirements.

Author Response

Itemized Replies to Reviewer’s Comments

Re: manuscript “Feasibility study of activated sludge/contact aeration combined system treating oil-containing domestic sewage” by Chen et al. (Manuscript ID: ijerph-638857)

Thank you very much. We appreciated for your comments.

Revise 1(round2)

The writing of Introduction is both abnormal and too simple. It doesn’t provide sufficient background introduction of the related studies.

A:Although traditional biological treatment has the advantage of low operation and maintenance costs, it requires a large place for facilities, and the AS method can be utilized to treat up to only 30mg/L of oil and grease (Razaviarani et al., 2013). Reduction of oil and grease may be due to biological growth that destroys the emulsifying agent, which has sufficient adsorptive power to hold oil and grease for its oxidation (Bishnoi et al., 2006).

Doesn’t clearly note the meaning of this study.

A:The meaning of this is Titel: Feasibility study of activated sludge/contact aeration combined system treating oil-containing domestic sewage.

The number of significant digits for the effective volume in Table 5 and the texts should be unified.

A:

Table 5. Size and Effective Volume of Each Tank in Small Model Factory.

Item

Dimensions

L×W×H(cm)

Effective Volume

(liter)

No. of Tanks

Activated Sludge Tank

15×15×20

4.50

2

Contact Aeration Tank

15×15×20

4.50

2

Sediment Tank Square Upper Half

15×15×10

2.25

2

Conical Lower Half

15×15×8

0.58

2

Line 69-76: The running parameters cannot become sub-Sections. They could be changed to normal texts.

A: Controlled Conditions of Small Model Factory: Inflow Rate: 18.75 C.C./min. Hydraulic Retention Time (HRT) of Biological Aeration Tank is 8 hrs. HRT of Activated Sludge Tank is 4 hrs. HRT of Contact Aeration Tank is 4 hrs. HRT of Sedimentation Tank is 2.5 hrs.  Sludge Return Rate is 30 %. Sludge Retention Time (SRT)≒15 days. Filling Rate of Biological Contact Filter in CA Tank is 50%.

Table 2 – Table 6: Each inflow parameters’ values for all the lines should be provided in the tables.

A:

Table 2. Composition of Wei Lih Fragrance Oil

(Inflow parameters values are 15mg/L, 30mg/L, 60mg/L and 120mg/L)

Composition

Percentage (%)

Meat Coronene Acid

1.40~2.00

Palmitic Acid

18.05~20.98

Palmitoleic Acid

3.01~3.95

Stearic Acid

3.15~5.91

Oleic Acid

48.76~55.33

Linoleic Acid

13.20~18.31

α-Linoleic Acid

Trace

Table 3. Composition and Properties of Emulsifier

(Inflow parameters values are 1.5mg/L, 3.0mg/L, 6.0mg/L and 12.0mg/L)

1. Brand and Product Number: Sigma-Aldrich 274348

2. Chemical Name: Polyethylene glycol sorbitan monolaurate

3. Average Mol. Wt.: 1,228

4. Composition: Lauric acid, 50%

(with balance primarily myristic, palmitic, and setaric acids)

5. Fp > 230℉

6. Density: 1.095 g/mL at 25℃

Table 4. Specifications of Biological Contact Filter in Model Factory

(Filling rate of biological contact filter in CA tank is 50%)

Model

Material

Size

D×H

(mm)

Density

(kg/m3)

Specific Surface Area

(m2/m3)

Porosity

(%)

Number of per Unit Volume

(pieces/m3)

Double

Star ball

PVC

32×27

76

274

91

74,000

Table 6. BOD of Synthetic Domestic Sewage Treated Using AS System and AS/CA System.

Period

No. of Days

BOD-in

(mg/L)

AS system

AS/CA system

BOD-out (mg/L)

 Removal Rate
(%)

BOD-out (mg/L)

 Removal Rate
(%)

Stable period

29

212

5

97.64

3

98.58

33

203

3

98.52

2

99.01

(14 days)

40

224

7

96.88

3

98.66

Mean Average

213

5

97.65

3

98.59

The appearances of the tables and the figures are ugly. Their format should be uniformed according to the journal requirements.

A:

Table 1. Composition of Synthetic Domestic Sewage

Sewage Composition

Concentration (mg/L)

Glucose

400

Nutritious Beef Juice

49

Sodium Carbonate

(Depend on pH value)

Iron Chloride

400

Ammonium Chloride

49

Dipotassium Hydrogen Phosphate

12

Sodium Sulfite

38

Average COD

14

Average BOD

17

Table 2. Composition of Wei Lih Fragrance Oil

(Inflow parameters values are 15mg/L, 30mg/L, 60mg/L and 120mg/L)

Composition

Percentage (%)

Meat Coronene Acid

1.40~2.00

Palmitic Acid

18.05~20.98

Palmitoleic Acid

3.01~3.95

Stearic Acid

3.15~5.91

Oleic Acid

48.76~55.33

Linoleic Acid

13.20~18.31

α-Linoleic Acid

Trace

Table 3. Composition and Properties of Emulsifier

(Inflow parameters values are 1.5mg/L, 3.0mg/L, 6.0mg/L and 12.0mg/L)

1. Brand and Product Number: Sigma-Aldrich 274348

2. Chemical Name: Polyethylene glycol sorbitan monolaurate

3. Average Mol. Wt.: 1,228

4. Composition: Lauric acid, 50%

(with balance primarily myristic, palmitic, and setaric acids)

5. Fp > 230℉

6. Density: 1.095 g/mL at 25℃

Table 4. Specifications of Biological Contact Filter in Model Factory

(Filling rate of biological contact filter in CA tank is 50%)

Model

Material

Size

D×H

(mm)

Density

(kg/m3)

Specific Surface Area

(m2/m3)

Porosity

(%)

Number of per Unit Volume

(pieces/m3)

Double

Star ball

PVC

32×27

76

274

91

74,000

Table 5. Size and Effective Volume of Each Tank in Small Model Factory.

Item

Dimensions

L×W×H(cm)

Effective Volume

(liter)

No. of Tanks

Activated Sludge Tank

15×15×20

4.50

2

Contact Aeration Tank

15×15×20

4.50

2

Sediment Tank Square Upper Half

15×15×10

2.25

2

Conical Lower Half

15×15×8

0.58

2

AS system

AS/CA system

Figure 1. Flow Chart of Small Model Factory with AS and AS/CA Systems.

Figure 2. COD of Synthetic Domestic Sewage Treated by AS and AS/CA Systems.

Reviewer 4 Report

The analytical methods, the standard methods for the examination of water and wastewater, should be written in the manuscript and references. 

Are there no references regarding composition of sewage and oil in Table1 to 2?

Please provide a photo of biological contact filter.

Author Response

Itemized Replies to Reviewer’s Comments

Re: manuscript “Feasibility study of activated sludge/contact aeration combined system treating oil-containing domestic sewage” by Chen et al. (Manuscript ID: ijerph-638857)

Thank you very much. We appreciated for your comments.

Revise 4(round2)

The analytical methods, the standard methods for the examination of water and wastewater, should be written in the manuscript and references.

A:The analytical methods based on the standard methods for the examination of water and wastewater.

Are there no references regarding composition of sewage and oil in Table1 to 2?

A:Table 1 and Table 2 based on “Chen,C.K and Lo,S.L.,2006. The study of treatment functions of activated sludge/contact aeration combined system. PHD paper of Graduate Institute of Environmental Engineering National Taiwan University” and paper of Development Center for Biotechnology in Taiwan.

Please provide a photo of biological contact filter.

A:
